# Territorial differences in the spread of COVID-19 in European regions and US counties

**Fabrizio Natale***, Stefano Maria Iacus, Alessandra Conte, Spyridon Spyratos, Francesco Sermi

European Commission - Joint Research Centre - Ispra, Ispra, Italy

* Fabrizio.NATALE@ec.europa.eu

## Abstract

This article explores the territorial differences in the onset and spread of COVID-19 and the excess mortality associated with the pandemic, with a focus on European regions and US counties. Both in Europe and in the US, the pandemic arrived earlier and recorded higher Rt values in urban regions than in intermediate and rural ones. A similar gap is also found in the data on excess mortality. In the weeks during the first phase of the pandemic, urban regions in EU countries experienced excess mortality of up to 68 pp more than rural ones. We show that, during the initial days of the pandemic, territorial differences in Rt by the degree of urbanisation can be largely explained by the level of internal, inbound and outbound mobility. The differences in the spread of COVID-19 by rural-urban typology and the role of mobility are less clear during the second wave. This could be linked to the fact that the infection is widespread across territories, to changes in mobility patterns during the summer period as well as to the different containment measures which reverse the link between mobility and Rt.

## Introduction

The COVID-19 pandemic is creating severe social and economic consequences, with some places experiencing disproportionately high levels of mortality and economic losses. Urban regions, and particularly large cities, have been severely affected by the spread of the pandemic in its early stages. Public discussion on the territorial impact of the pandemic requires a greater understanding of the way the pandemic is affecting regions that are diversely vulnerable and will require different recovery plans. Analyses of the role of population density and city size on the virus spread have led to mixed results [1–3]. While these analyses primarily look at the population scale as a whole, other analyses have examined disparities within the urban environment, looking in particular at the intensity of social contacts related to the urban organisation and life that would make some places more prone to infection in the first phase. In particular, some of the factors considered relevant for virus transmission are the connectivity of cities as hubs of national and international transport systems [4–6], and the structure of industry and the concentration of essential jobs in certain areas. [7, 8]. In addition, it has been documented that COVID-19 transmission is highest in family environments where people

**Data Availability Statement:** All data used in this study, except the mobility data, are openly available, and the data sources are specified in the data and methods section of the manuscript. Mobility data cannot be shared publicly because of

legal reasons. We made publicly available all data used for the Table1 and Table2 as well as the R code to produce the nine models presented in the respective tables in the SI. We added a random noise (non negative and skewed) to the mobility data since due to their commercial sensitivity we cannot published them. Interested researchers who wish to reproduce the regressions with mobility variables can use openly available mobility data from other sources such as the Google mobility data. Future researchers can request access to the mobility data from the Mobile Network Operators who participate in the European Commission's initiative to fight the COVID-19 pandemic. More info about this cooperation agreement can be found on the "Letter of Intent for Cooperation" which is available at https://www.gsma.com/gsmaeurope/wp-content/uploads/2021/02/Letter-of-Intent_final_16-April-2021.pdf.

**Funding:** The authors received no specific funding for this work.

**Competing interests:** The authors have declared that no competing interests exist.

tend to be in close contact and with multi-generational family members living together [9, 10]. Our paper is aimed to gain a deeper understanding of the links between COVID-19, urban-rural typologies, territorial conditions, and mobility, which is critical for designing effective public health policy responses. We first explore the heterogeneity of COVID-19 patterns in its onset, spread, and associated excess mortality by comparing the results by the level of urbanisation of European regions and counties in the US. For the EU we use Eurostat NUTS3 rural-urban typologies and for the US we use the Rural-Urban Continuum Codes reduced to 3 classes. The classification in the EU and the US according to rural-urban typologies follows harmonised criteria of population density and size of the urban centres. On the basis of the share of the rural population regions at Territorial Level 3 (i.e. NUTS3 in the EU and counties in the US) are classified as predominantly rural regions, intermediate regions and predominantly urban regions. These classifications are routinely used by National Statistical Offices, the OECD and by the European Commission to publish territorial statistics. The results of our comparison of the spread of COVID-19 across regions show that the pandemic started earlier in urban regions than in intermediate and rural ones. Urban regions had the highest Rt values in both Europe and the US during the first wave, whereas rural counties were more affected than urban counties in the second wave. Analysis of excess mortality, calculated using Eurostat statistics and obtained from the difference between reported fatalities and a baseline model based on historical data between 2011 and 2019, also shows a large gap by urbanisation level during the first wave, with a median excess mortality up to 73% for urban regions, 18% for intermediate regions, and 11% for rural regions. In a second phase, we empirically examine the impact of mobility on virus spread. We model population mobility in European regions through a harmonised mobility index derived from mobile phone data. For our purpose of comparison by rural-urban typologies these data is unique because it provides not only relative temporal variation of mobility within each region in respect to a reference date but also information about absolute differences across regions. Due to the lack of similar data for the US our analysis on the effect of mobility on Rt is limited to the EU. We examine the geographical distribution of mobility changes through regression models for the weeks in the first and second virus waves. Our results show that, on the one hand, higher mobility explains most of the variation in values in the weekly Rt during the first wave, with internal, inbound, and outbound mobility positively affecting Rt. The effect of the per capita internal mobility, in particular, is more pronounced than that of the degree of urbanisation, and remains significant even when population and population density are taken into account. On the other hand, the same regression models replicated for the second wave show a negative role of mobility on the local spread of the virus, as well as a higher prevalence of the infection in rural regions compared to large cities. The paper is organised as follows. The data section describes the data and methods used in the analyses. In the results and discussion section, we present how the COVID-19 pandemic spread in rural, intermediate, and urban regions during the first and the second wave, and the conclusions are outlined in the final section.

## Data and methods

In this section we present the data sources and the methods we used to assess the spread of the COVID-19 pandemic in rural, intermediate and urban regions. EU regions are classified into three types of areas based on the share of the local population living in urban clusters and city centres: urban (densely populated areas), intermediate (intermediate density areas), and rural areas (sparsely populated areas) (For more details on the Eurostat classification, see the link https://ec.europa.eu/eurostat/web/degree-of-urbanisation/background. For a general overview of the different approaches to the delineation of a city, see Rozenblat C.,(2020). US counties

follow a similar classification scheme that distinguishes metropolitan counties by the population size of their metropolitan area and non-metropolitan counties by the degree of urbanisation and their proximity to a metropolitan area see https://www.ers.usda.gov/data-products/rural-urban-continuum-codes.aspx.) In this analysis, the variable grouping the 2013 rural-urban codes has been reclassified into 3 categories: urban (codes 1–2), intermediate (codes 3–4), and rural counties (codes 5–9). We calculated the reproductive number (Rt) as an indicator to assess how fast the virus spread across different types of geographical areas. We estimated the excess mortality to monitor in quantitative terms the evolution and impacts of COVID-19 pandemic. We used fully anonymised and regionally aggregated mobility data to get insights about the different regional mobility patters. Finally, we fitted a linear regression model to assess the relationship between mobility and Rt during the first and the second wave.

## Rt

Rt is the main real-time indicator used to assess the evolution of the pandemic, design containment measures and monitor their effectiveness. During the pandemic several governments and administrations have established systems for the automatic triggering of restriction measures based on a weekly monitoring of regional Rt values. Technically, Rt gives a measure of the number of new infections caused by infected individuals at time t in a partially susceptible population. Values above one indicate that that the number of cases will increase while with values below one the pandemic will extinguish. A time-dependent reproduction number, Rt, was calculated for each day and region with the R package 'R0' [11]. For the calculation we followed a likelihood-based estimation procedure that derives the probability of infection from the analysis of the epidemic curve of the observed cases using sliding temporal windows. [12] This estimation procedure relies on a parameter about the time between the infection and the manifestation of the symptoms which in our cases was obtained from data reported during the early phases of the pandemic in China [13]. The data on confirmed COVID-19 cases at regional level was obtained through the 'COVID19' R package [14] and updated until end of 2020. To analyse at descriptive level territorial differences, the daily Rt values were averaged by consecutive weeks and across regions classified according to their rural-urban typology.

## Excess mortality

The baseline for mortality was calculated with Generalised Additive Models fitted independently for each region. In the models we included a seasonal component to account for the increase in mortality during the winter months linked to influenza outbreaks, and a linear time trend to account for long-term changes in mortality due to demographic dynamics. The excess mortality was measured as difference between the reported data in 2020 and the estimated baseline for all occurrences exceeding the lower or upper 95% confidence intervals of the estimated baseline. The weekly mortality were obtained from Eurostat (demormweek3) and covered 900 regions in 26 EU Member States and the UK with time series which, depending on the MS, were starting from 2001 or 2015 and spanning until the end of 2020.

## Mobility

In this study we used fully anonymised and aggregated mobility data shared with the European Commission (EC) by European Mobile Network Operators (MNOs). These mobility data comply with the 'Guidelines on the use of location data and contact tracing tools in the context of the COVID-19 outbreak' by the European Data Protection Board [15]. The mobility data were in the form of Origin-Destination Matrix (ODM) [16, 17] and they provided valuable insights into mobility patterns across geographical areas. The data has been used to derive

mobility insights and build tools to inform better targeted containment measures, in a Mobility Visualisation Platform, available to the Member States [18].

Given the high variation in the spatial and temporal aggregation across countries and operators, the original ODMs were harmonised at standardised spatial and temporal granularity to the derived Mobility Indicators [19]. We further aggregated the Mobility Indicators at weekly intervals, and we normalised the Mobility Indicators to enable a better cross-country comparison. The normalisation was performed by comparing the number of movements for each NUTS3 areas and each type of movements (internal, inbound, outbound) by the average mobility levels between February 10 and March 8, 2020. The reason for this normalisation was to capture the relative decrease/increase of mobility compared to pre-lockdown levels. In addition to normalised mobility, we also estimated the per-capita internal mobility by dividing the number of movements recorded using mobility data in a NUTS3 region by population size reported by Eurostat as of 1 January 2018. The number of movements recorded by each Mobile Network operator depends on their methodology and their penetration rate in each country. Thus to enable cross-country comparison, we normalised the per-capita internal mobility by setting for each country the value of one to the NUTS3 regions with the higher per capita mobility over the reference time period February to December 2020, and the value of zero to the NUTS3 regions with the lowest per-capita mobility over the same time period. The limitation of our proposed indicators are the following. First we assume that the penetration rate of each MNO in each country is the same across rural, intermediate and urban areas and it remains stable across the time period that we analyse. Second, we assume that the population of the NUTS3 areas remained stable during the period that we analyse.

## Regression

To support our intuition about the territorial heterogeneity in the spread of COVID-19 during the first and second waves, we examine the effect of different mobility patterns through OLS regression models. The models have the Rt values recorded in each European region as dependent variable, the rural-urban typology of the region, the internal, outbound and internal per capita mobility as main independent variables and the logs of the population and population density of the region as control variables. We run two set of models at 28 days since the onset of the pandemic in each region to capture effects during the first wave and for the weeks after August 2020 for the second wave. All specifications include country fixed effects to account for differences in virus transmission resulting from invariant country characteristics. The fitting of the regression models was constrained by the necessity of having regional data on COVID-19 cases for the calculation of Rt and mobility indicators for the same periods. Data on population was obtained from Eurostat (demorpjangrp3 and demord3dens). Overall the regressions are based on around 3500 observations in 654 regions for the first wave, and 10500 observations in 551 regions for the second wave.

## Results and discussion

### The COVID-19 pandemic started earlier in urban regions

Fig 1 describes the pandemic onset in the NUTS3 regions of some European countries and counties in the US, clustered by the three levels of urbanisation. We measure the pandemic onset in each region by the number of days between the registration of the first 20 confirmed cases of Coronavirus disease and the beginning of the year 2020. In both Europe and the United States, urban regions are more vulnerable to the pandemic's onset. The pandemic started earlier in most urban regions, while we observe a later onset in intermediate and rural regions in the first wave. The choice of the threshold of 20 cases was to avoid influences from

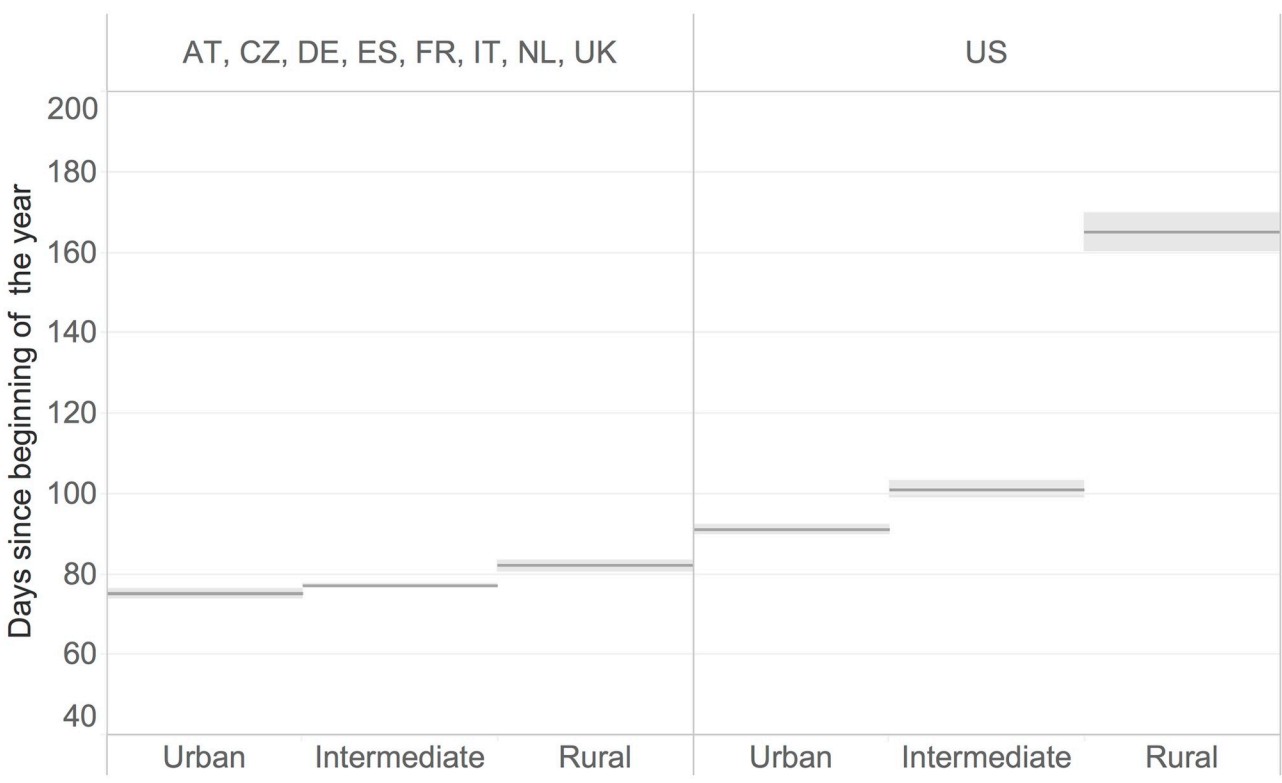

**Fig 1. Onset of the pandemic across regions by rural-urban typology.**

sporadic events. We performed a sensitivity check using thresholds between 1 and 500 cases and the finding of an earlier onset in urban regions in respect of intermediate and rural ones holds for all values of the threshold.

## The infection has spread faster in urban regions during the first wave

Fig 2 displays the Rt values for the first and second waves of the pandemic. Rt is calculated from daily confirmed cases in 807 NUTS3 regions in the UK, Netherlands, Germany, Italy, Spain, France, Czech Republic and Austria (left) and in 3100 counties in the US (right). The indicator is averaged across regions grouped by rural-urban type and aggregated by days since the first reported case in each region (upper Figure), and weeks since the start of the second wave of the pandemic (lower Figure). Looking first at the upper figure, we observe that urban regions in Europe and the US recorded higher Rt values than those found in intermediate and rural regions at the start of the pandemic. This indicates that the disease spread faster in urban regions and that containment was more difficult in more densely populated areas. Approximately 56 days after the start of the pandemic, we find a general decline in the Rt and a reduction in the differences in Rt between the three groups of regions. At the start of the pandemic, the rural-urban divide in Rt values is more pronounced in the US counties. However, even in this case, the disparity in the pandemic spread by level of urbanisation has narrowed among the three regional groups, with the Rt index close to 1 at the end of the first wave. The lower part of Fig 2 shows the median Rt values across regions and counties in the weeks following the summer period, when the pandemic began to spread in a second wave of infections. In the European regions, we observe an initially higher Rt in the urban regions and increasing and

First wave (days since onset)

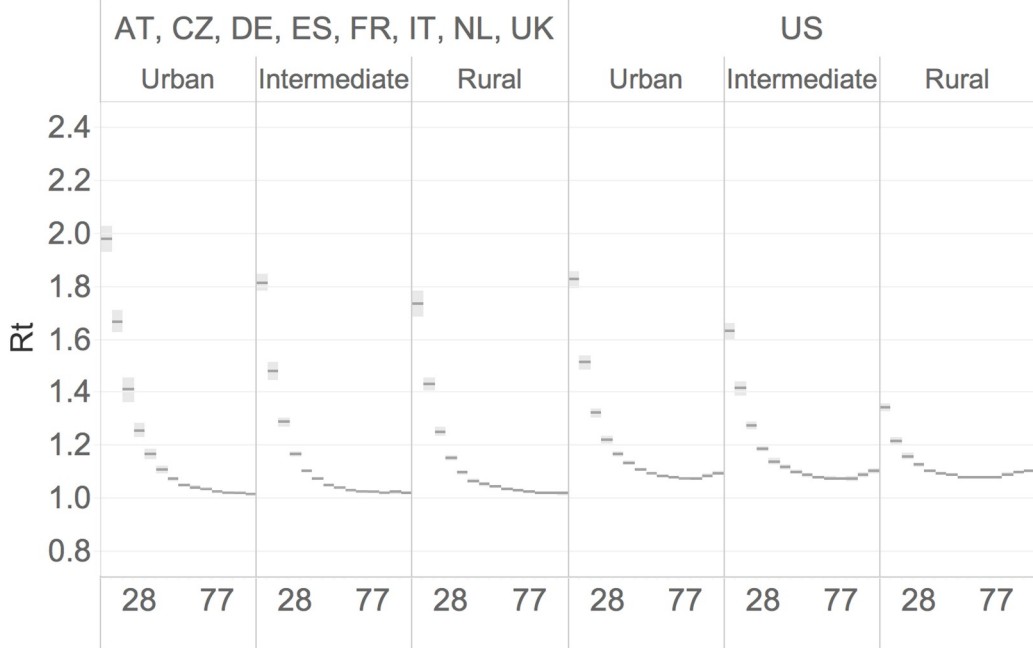

Second wave (calendar weeks)

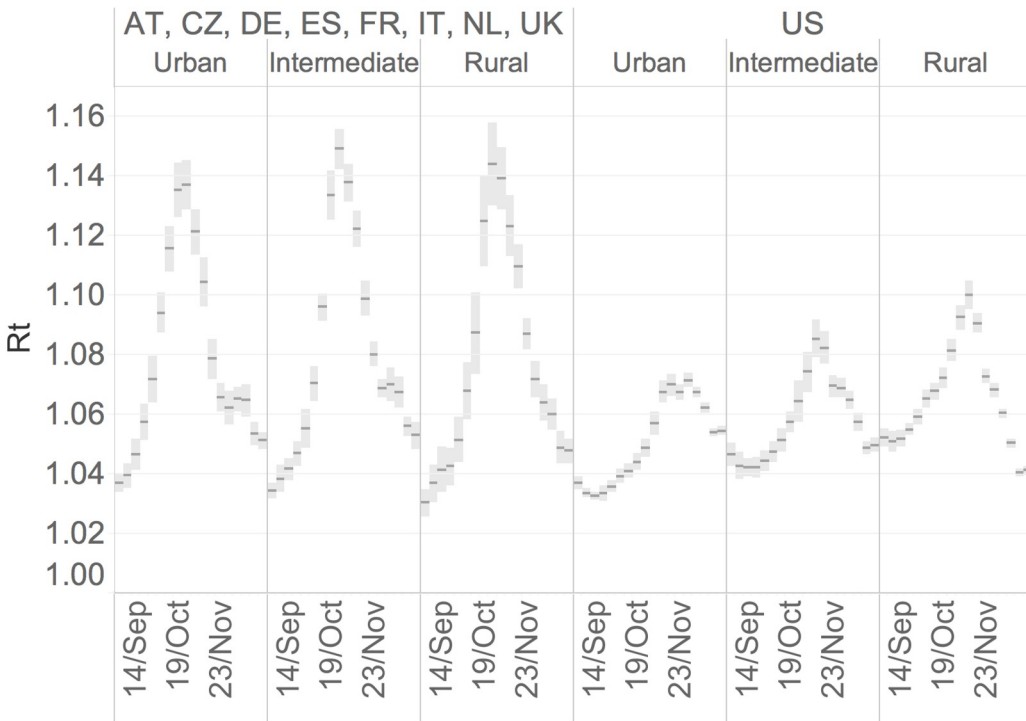

**Fig 2. Median Rt values across European regions and US counties by rural-urban typology and sliding temporal windows.** The upper part presents the median Rt values for days from the first recorded case in each region during the first wave. The lower part represents the median Rt values for calendar weeks during the second wave.

higher values in the intermediate and rural regions as the second wave progresses. In contrast, in the US, rural and intermediate counties are the most vulnerable to virus spread for most of the weeks during the second wave, with a slight change in trend in the last weeks of the period.

## The excess mortality linked to COVID-19 is higher in the European urban regions in the first wave

Fig 3 shows the trend in the excess mortality for the European regions during the year 2020. The increase in weekly mortality compared to past trends is used as an indirect measure to monitor the evolution of COVID-19. This indicator has the downside of including fatalities not necessarily linked to COVID-19, such as those caused by the saturation of hospital capacity, but has the advantage of being less influenced by the underestimation of the real infection rate due to asymptomatic cases or differences in testing strategies over time and regions [20]. The bars in Fig 3 show the weekly total excess mortality calculated from Eurostat statistics for most EU countries and the UK. The excess mortality is obtained from the difference between the reported fatalities and a modelled baseline estimated from historical data until 2019. The number of weekly fatalities attributable to COVID-19 peaked at the beginning of April, with about 41 400 deaths in excess compared to the baseline.(This peak represents 21 600 more cases than the excess mortality recorded in the same countries during the second week of January 2017, corresponding to a particularly severe year for the seasonal flu.) The lines in the figure show the median excess mortality in the NUTS3 regions classified according to their degree of urbanisation. At the peak of the pandemic in third week of April, the median excess mortality in urban regions reached its peak with an excess mortality of 73%, which was 58 pp higher than in intermediate regions and 68 pp higher than in rural regions in the same week.

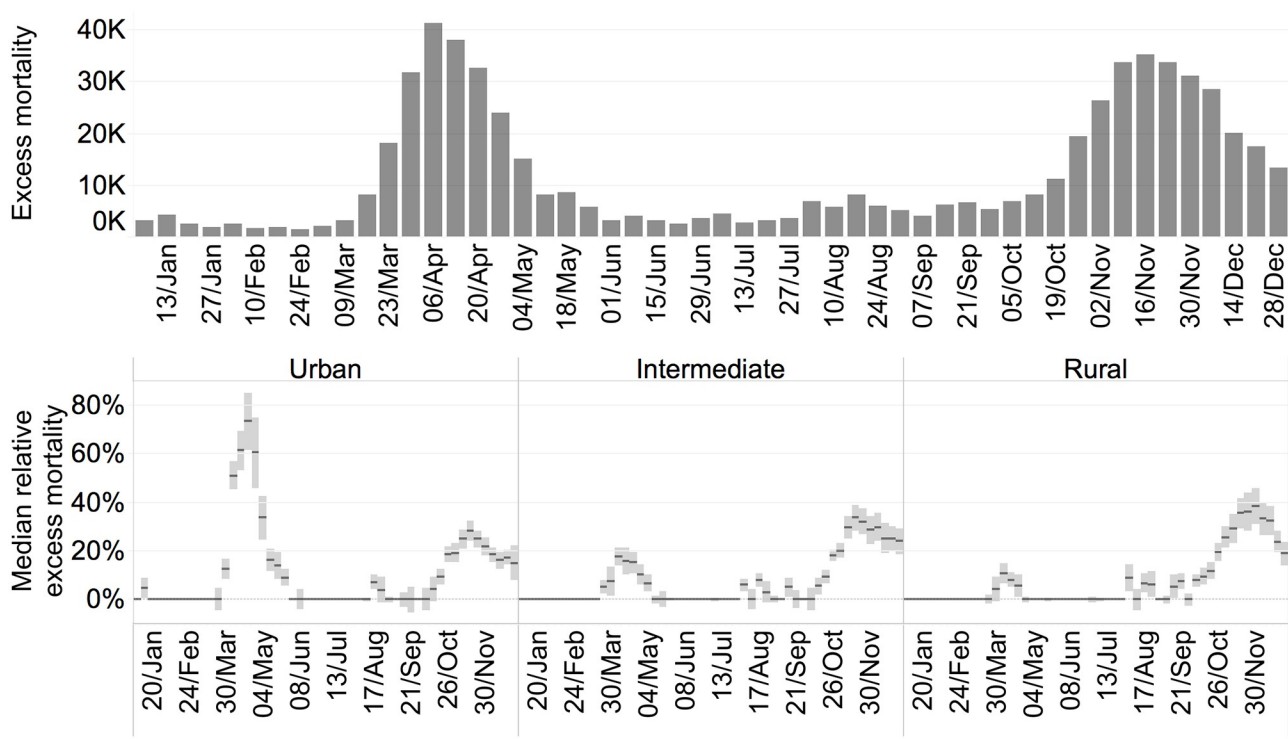

**Fig 3. Total excess mortality (upper panel) and median relative excess mortality with 95% CI by rural-urban typology (lower panel) on a weekly basis.**

In the second wave of the pandemic, the disparities among regions appear less pronounced. There is also a reverse in the trend of excess mortality, with rural and intermediate regions having higher rates, 38% and 32% respectively, than urban regions with an excess mortality rate of 26%.

## Mobility is higher in urban regions

One possible explanation for the higher Rt and excess mortality in urban regions is that in cities the infection can spread more rapidly given the higher population density, larger use of public transportation and higher number of social interactions. The intensity of social interaction is reflected in mobility indicators which can be calculated from mobile phones data. In fact, the relation between intensity of social contacts, mobility and infection is at the basis of mobility restriction that most governments have put in place to contain the pandemic. We analyse the patterns of mobility within, from and toward European regions with anonymised and aggregated mobile indicators derived from mobile phone data as described in the Data and methods section. Fig 4 shows the median patterns of weekly mobility of 1033 NUTS3 regions in 22 EU countries, grouped by rural, intermediate and urban typology, in relative terms compared to the pre-lockdown levels (upper chart) and per capita terms (lower chart). The trends in the two charts in Fig 4 reflect the implementation of generalised lockdown until April, the reopening during the summer period and the new restrictions on mobility after summer. The upper chart in Fig 4 reflects that the mobility has been reduced more in relative terms in urban areas compared to the intermediate and rural ones. The lower chart in Fig 4 shows that during the first wave, and independently from the implementation of the restriction

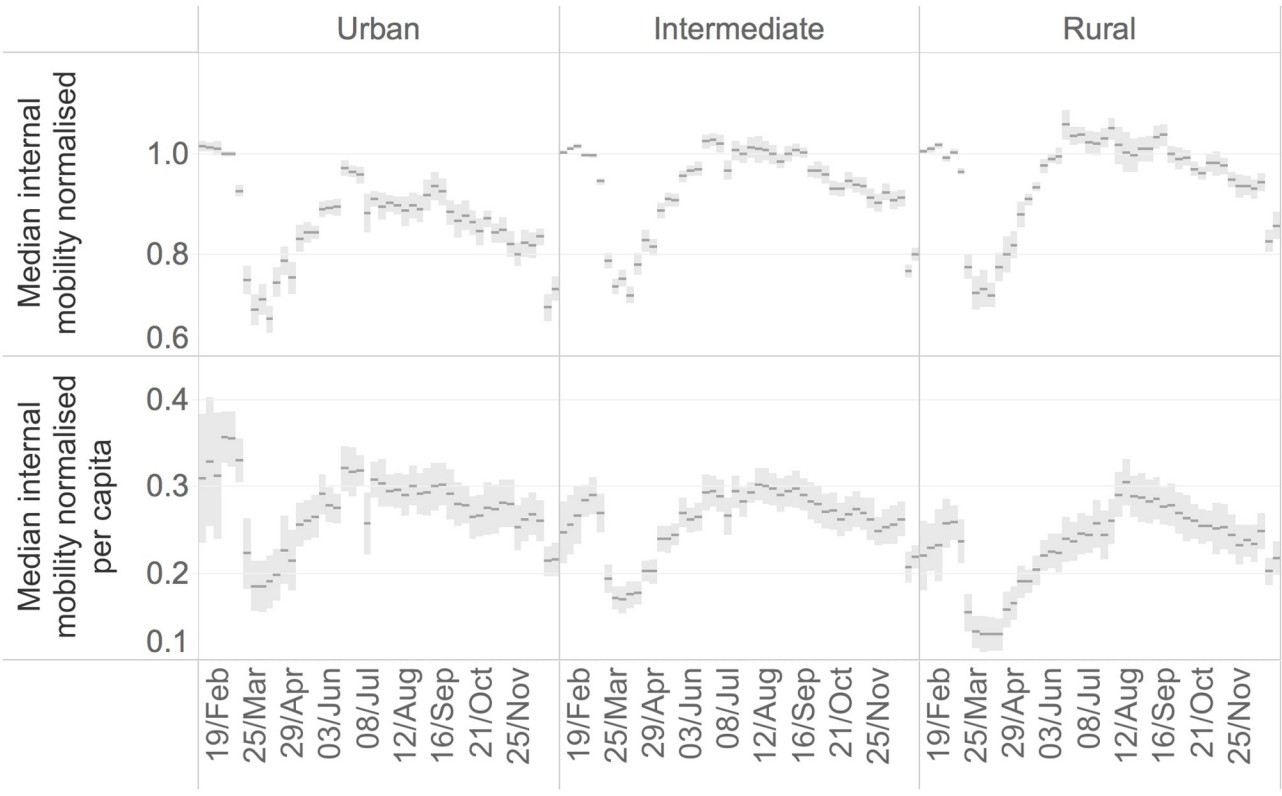

**Fig 4. Median normalised internal mobility and internal mobility per capita with 95% CI, across regions grouped by rural-urban typology.**

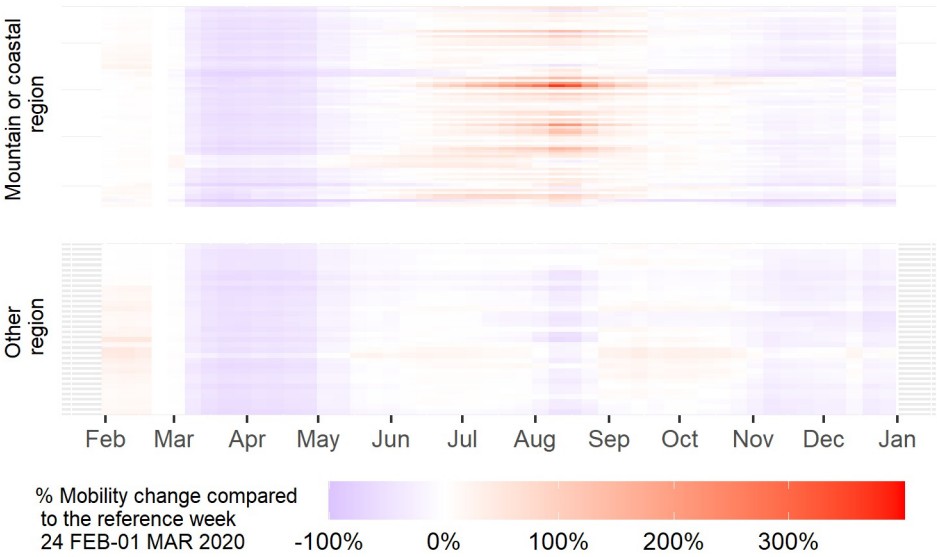

**Fig 5. Percentage weekly relative change in the internal mobility of NUTS3 areas in Italy in respect to the pre-COVID-19 mobility levels (week 24 FEB–01 MAR 2020).**

measures, the level of per capita mobility was higher in urban regions in respect of intermediate and rural ones. During the second wave, the per capita mobility is almost equal across all areas, indicating substantial reduction of mobility in urban and intermediate regions at the beginning of summer and an increase in rural regions. This shift in mobility patterns is exemplified in Fig 5 showing the weekly relative changes in mobility for each Italian region (rows) in respect of the levels recorded during the last week of February. In this case, regions are sorted on the basis of their proximity to the sea or mountains to better appreciate the mobility linked to domestic tourism. In May, after the lifting of lockdown, all Italian regions recorded an increase of mobility to the levels of February. However, during summer, in coastal and mountain regions mobility increased to higher values than at the beginning of the year. The highest increase was recorded in the second week of August in the renowned region of Olbia in Sardinia (+373%). The fact that there was high mobility from urban to coastal and mountainous areas could have contributed to spreading the disease from cities to intermediate and rural areas. With the re-opening of schools in September, the level of mobility started again to increase uniformly across all regions.

## The higher mobility in urban regions may explain great part of the territorial gaps in Rt during the first wave

Table 1 shows the results of regressions on the first wave of infection considering Rt values in the 28 days after the start of the pandemic. Table 2 presents results for the second wave on the Rt values in the weeks after August. The results of the regressions show a significant relationship between the effective reproduction number, Rt, and the levels of urbanisation (Column 1 in Table 1). During the first wave of the pandemic, Rt values are lower in rural and intermediate regions than in the urban regions used as reference. Urban regions are therefore the most affected in the first weeks of the pandemic in terms of number of cases due to their high population density and large concentration of social interactions, as well as the high local and global connectivity (Balcan et al., 2009). In Columns 2–4 we include the mobility controls separately,

**Table 1. Regression on Rt during the first wave (28 days since onset).**

| | Rt first wave | | | | | | | | |
|---|---|---|---|---|---|---|---|---|---|
| | **(1)** | **(2)** | **(3)** | **(4)** | **(5)** | **(6)** | **(7)** | **(8)** | **(9)** |
| Intermediate | −0.13** | | | | | | | 0.03 | 0.05 |
| | (0.05) | | | | | | | (0.03) | (0.04) |
| Rural | −0.17*** | | | | | | | 0.06* | 0.08 |
| | (0.06) | | | | | | | (0.03) | (0.05) |
| Internal -3W | | 1.82*** | | | | | | | |
| | | (0.07) | | | | | | | |
| Inbound -3W | | | 1.52*** | | | | | | |
| | | | (0.05) | | | | | | |
| Outbound—3W | | | | 1.53*** | | | | | |
| | | | | (0.05) | | | | | |
| Internal pca -3W | | | | | 0.85*** | | | 0.89*** | 0.95*** |
| | | | | | (0.09) | | | (0.09) | (0.10) |
| log(Population) | | | | | | 0.11*** | | | 0.06*** |
| | | | | | | (0.03) | | | (0.02) |
| log(Population density) | | | | | | | 0.05*** | | −0.02 |
| | | | | | | | (0.02) | | (0.01) |
| AIC | 11126.3 | 5134.8 | 4933.1 | 4941.3 | 5715.6 | 11118.2 | 11124.6 | 5716.7 | 5711 |
| Observations | 3,563 | 3,025 | 3,025 | 3,025 | 3,025 | 3,563 | 3,563 | 3,025 | 3,025 |
| $R^2$ | 0.02 | 0.31 | 0.35 | 0.35 | 0.16 | 0.02 | 0.02 | 0.16 | 0.17 |
| Adjusted $R^2$ | 0.01 | 0.31 | 0.35 | 0.35 | 0.16 | 0.02 | 0.01 | 0.16 | 0.16 |
| F Statistic | 7.42*** | 192.71*** | 235.70*** | 233.89*** | 83.74*** | 9.37*** | 8.44*** | 65.47*** | 54.58*** |
| df | 8; 3554 | 7; 3017 | 7; 3017 | 7; 3017 | 7; 3017 | 7; 3555 | 7; 3555 | 9; 3015 | 11; 3013 |

*Note*:

*$p < 0.1$;

**$p < 0.05$;

***$p < 0.01$

Standard errors in parenthesis

i.e. internal, inbound, outbound mobility, given the correlation between these measures within countries. We use the three-week lagged value of each mobility variable in the regressions to account for the delay between the mobility-driven infection and the positive case confirmation and to mitigate a potential reverse causality problem between the two variables. A sensitivity checks for the choice of the alternative lag periods is shown in Fig 6. We selected a lag of 3 weeks which is maximising the positive coefficient of mobility during the second wave. Positive lags produce as expected negative coefficients since mobility is reacting to restrictions measures rather than driving the infection. In all specifications, each mobility indicator is positively correlated with Rt values, indicating that higher mobility is associated with higher transmission. The coefficient on delayed mobility ranges from 1.82 to 1.53, depending on the specification. Mobility is also analysed using a per capita mobility indicator (Column 5), which captures the daily movements per capita in a nuts region. The positive and significant coefficient of the per capita mobility confirms a pattern of Rt that increases as the internal mobility measured on the total population increases. The demographic controls of the (log) total population and density, presented in Columns 6 and 7, also exert a positive effect on Rt in the first wave. Finally, in Columns 8 and 9, we simultaneously estimate the effect of the per capita

**Table 2. Regression on Rt during the second wave (after August).**

| | Rt second wave | | | | | | | | |
|---|---|---|---|---|---|---|---|---|---|
| | (1) | (2) | (3) | (4) | (5) | (6) | (7) | (8) | (9) |
| Intermediate | 0.004*** | | | | | | | 0.004*** | −0.004** |
| | (0.001) | | | | | | | (0.001) | (0.002) |
| Rural | 0.01*** | | | | | | | 0.01*** | −0.001 |
| | (0.002) | | | | | | | (0.002) | (0.002) |
| Internal -3W | | 0.02*** | | | | | | | |
| | | (0.003) | | | | | | | |
| Inbound -3W | | | 0.01*** | | | | | | |
| | | | (0.001) | | | | | | |
| Outbound—3W | | | | 0.01*** | | | | | |
| | | | | (0.001) | | | | | |
| Internal pca -3W | | | | | −0.03*** | | | −0.03*** | −0.03*** |
| | | | | | (0.01) | | | (0.01) | (0.01) |
| log(Population) | | | | | | −0.01*** | | | −0.01*** |
| | | | | | | (0.001) | | | (0.001) |
| log(Population density) | | | | | | | −0.004*** | | −0.002*** |
| | | | | | | | (0.001) | | (0.001) |
| AIC | -29920.5 | -29875.7 | -29856.1 | -29871.3 | -29884.7 | -29957.2 | -29933.6 | -29934.3 | -30013.5 |
| Observations | 10,720 | 10,716 | 10,711 | 10,713 | 10,717 | 10,720 | 10,720 | 10,717 | 10,717 |
| $R^2$ | 0.07 | 0.06 | 0.06 | 0.06 | 0.06 | 0.07 | 0.07 | 0.07 | 0.08 |
| Adjusted $R^2$ | 0.07 | 0.06 | 0.06 | 0.06 | 0.06 | 0.07 | 0.07 | 0.07 | 0.08 |
| F Statistic | 95.50*** | 103.95*** | 102.92*** | 104.58*** | 104.82*** | 114.48*** | 110.86*** | 87.88*** | 80.04*** |
| df | 8; 10711 | 7; 10708 | 7; 10703 | 7; 10705 | 7; 10709 | 7; 10712 | 7; 10712 | 9; 10707 | 11; 10705 |

*Note*:

*p<0.1;

**p<0.05;

***p<0.01

Standard errors in parenthesis

internal mobility, the level of urbanisation of the regions and the population density and size. The main result of the estimates is that the increase in the internal mobility is positively and significantly associated with the number of cases, with a stable coefficient across the different specifications. The coefficient of the internal mobility indeed remains significant and positive even when we include the other control variables. Internal mobility appears to be a critical determinant of the rate of COVID-19 cases during the first wave, positively influencing the spread of the virus possibly through increased social interactions. These results confirm that great part of the territorial characteristics influencing the higher epidemiological risk at the onset of the pandemic in urban regions can be explained by the role of mobility.

Table 2 examines the relationship between Rt and different mobility patterns in the European regions in a similar way to Table 1 but with data for the second wave (from August). The estimates show an inversion of sign from the first wave with a positive association between the virus spread and the rural and intermediate regions compared to large cities. These results may reflect a behavioural response as well as more severe containment measures in the most severely affected areas. In the second wave, different mobility patterns are associated with lower Rt values, presenting a weaker relationship than in the first wave, as

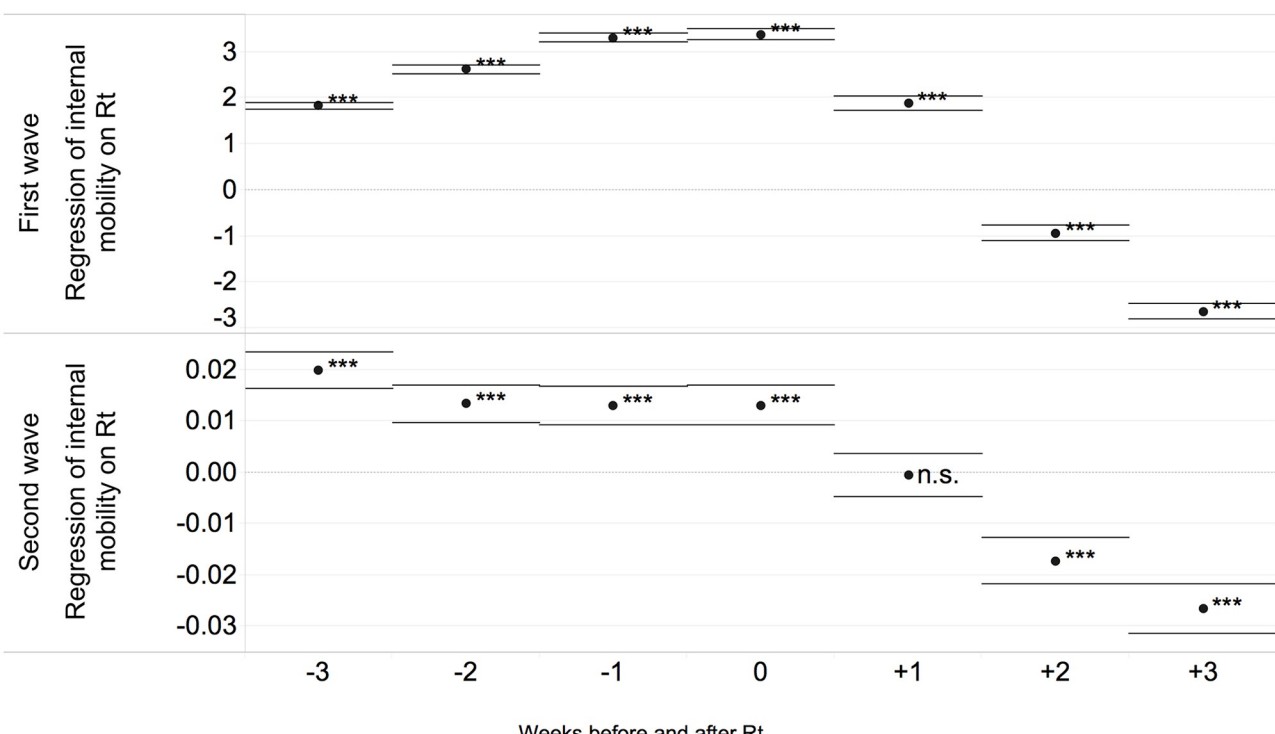

**Fig 6. Regression coefficients with shifts of internal mobility of 3 weeks before and after the reference week of Rt.**

presented in Column 2–6. The results thus indicate that the relationship between mobility and the regional virus transmission has changed over time and that shifts in mobility were used to control the pandemic. However, these changes were not sufficient to prevent a second wave of infection in most of the regions analysed. The demographic variables are significant and negative on the virus spread. The models that simultaneously estimate the effect of internal mobility per capita and different regional characteristics show a negative relationship between this mobility pattern and the virus transmission, as well as a higher prevalence of the infection in rural regions compared to large cities during the second wave. Fig 6 show the regression coefficients with internal mobility shifts of 3–0 weeks before and after the Rt reference week. This study is not aimed at a causality analysis between the two variables, however we quantify the different time-lag effects to detect their potential influence on transmission, which is useful for the deasese monitoring policies. During the first wave the relation of mobility on Rt is positive and peaks during the same week (week 0). During the second wave, the relation is constantly decreasing towards negative values. The fact that the relation during the first wave is becoming clearer towards the reference week indicates that mobility is having an effect on Rt. In contrast, during the second wave there is an inversion in the relationship and mobility rather than influencing seem to react to changes in Rt by moving in opposite directions. Intuitively, this is in line with the consideration that during the advanced stages of the pandemic, mobility is highly conditioned by restriction measures and closures that are put in place in correspondence with increases in Rt. (A specification linking the disease values to mobility may suffer from reverse causality. To mitigate this potential problem, we use a three weeks lagged value of each mobility variable in the regressions.)

## Conclusion

In this article we analysed the territorial differences in the onset and spread of COVID-19 and the associated excess mortality, across the European NUTS3 regions and US counties during the first and second COVID-19 wave. During the first wave, the COVID-19 pandemic arrived earlier, recorded higher Rt values and had a higher impact in terms of excess mortality in urban regions compared to the intermediate and the rural ones. In the first wave, mobility influenced the spread of COVID-19, since the higher mobility of urban regions is explaining entirely the differences between the three groups of regions. The fact that these effects are more difficult to recognise in later stages of the pandemic can be tentatively explained by the widespread of the infection, the implementation of restriction measures which invert the link between mobility and Rt, often applied on a territorial basis, and the more complex mobility patterns experienced during the summer period.

Our findings are in line with previous studies identifying the role of mobility on virus spread in the early stages of the pandemic. To our knowledge, our research is unique in providing a broad geographic coverage and a high level of geographical detail, and in examining the role of regional mobility for the spread of COVID-19 through a unique data set derived from mobile phone data. In terms of policy implication, our research contributes to a better understanding of territorial characteristics of the spread of COVID-19, which is critical for designing effective public health policy responses, often decided at regional level.

## Supporting information

**S1 File. Data and R code used for the regressions presented in Tables 1 and 2.** The ZIP file contains the two data files "first_wave.csv" and "second_wave.csv", the r code "Script.R" and a read me file with the data description "data_ReadMe.txt". Please note that due to the commercial sensitivity of the mobility data, we have added a skewed non-negative random noise to the mobility columns, therefore the statistical results of the models that include mobility data are not replicable.
(ZIP)

## Acknowledgments

The authors acknowledge the support of European MNOs (among which 3 Group—part of CK Hutchison, A1 Telekom Austria Group, Altice Portugal, Deutsche Telekom, Orange, Proximus, TIM Telecom Italia, Tele2, Telefonica, Telenor, Telia Company and Vodafone) in providing access to aggregate and anonymised data. The authors would also like to acknowledge the GSMA (GSM Association of Mobile Network Operators.), colleagues from Eurostat and ECDC (European Centre for Disease Prevention and Control. An agency of the European Union.) for their input in drafting the data request.

Finally, the authors would also like to acknowledge the support from JRC colleagues, and in particular the E3 Unit, for setting up a secure environment to host and process of the data provided by MNOs, as well as the E6 Unit (the "Dynamic Data Hub team") for their valuable support in setting up the database.

## Author Contributions

**Conceptualization:** Fabrizio Natale.

**Data curation:** Fabrizio Natale, Stefano Maria Iacus, Spyridon Spyratos, Francesco Sermi.

**Formal analysis:** Fabrizio Natale, Stefano Maria Iacus, Alessandra Conte, Spyridon Spyratos, Francesco Sermi.

**Writing – original draft:** Fabrizio Natale, Stefano Maria Iacus, Alessandra Conte, Spyridon Spyratos, Francesco Sermi.

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
