## [Decision Letter · Decision Letter 0]

27 Apr 2021

PONE-D-21-07940

Territorial differences in the spread of COVID-19 in European regions and US counties

PLOS ONE

Dear Dr. Natale,

Thank you for submitting your manuscript to PLOS ONE. After careful consideration, we feel that it has merit but does not fully meet PLOS ONE’s publication criteria as it currently stands. Therefore, we invite you to submit a revised version of the manuscript that addresses the points raised during the review process.

The paper is of high interest and the two Reviewers suggest very specific improvements that are very relevant in my point of view and often easy to solve (most of them are minor revisions asking some precisions on some points). I hope that you will agree with these proposals and that we will be able to publish your paper soon.

I would add 3 points.

1) In the review 2, you will find the request to put more attention (small discussion) about urban/rural definition (point 10 of his review), it is essential to refer to the EU+UN-Habitat+OECD work on urban/rural delineations on which I guess you based your typology?

- Applying the Degree of Urbanisation — A methodological manual to define cities, towns and rural areas for international comparisons — 2021 edition, DOI: 10.2785/706535

A paragraph on this discussion would be appropriate as it is fundamental in your model.

For a comparison of different approaches of "what is a city?", I would suggest a synthesis that I made recently (But I would not want to oblige you to quote my own work):

- Rozenblat C. (2020). Extending the concept of city for the delineation of large urban regions (LUR) for the cities of the world, Cybergeo, https://doi.org/10.4000/cybergeo.35411

2) for the approach on mobility: I think that you should distinguish the "local mobility" that you consider, to the "global" connectedness that could be evaluated by the air passengers flows of airports. I think that urban areas are also previously affected because they are highly globally connected. For this discussion I woudl suggest:

- Balcan, D., Colizza, V., Gonçalves, B., Hu, H., Ramasco, J. J., & Vespignani, A. (2009). Multiscale mobility networks and the spatial spreading of infectious diseases. Proceedings of the National Academy of Sciences, 106(51), 21484-21489.

3) As a spreading infection, the discussion could interprete also (with some limits) the COVID diffusion processes from urban to semi-urban and rural areas.

We look forward to receiving your revised manuscript.

Kind regards,

Celine Rozenblat

Academic Editor

PLOS ONE

Additional Editor Comments:

The paper is of high interest and the two Reviewers suggest very specific improvements that are very relevant in my point of view and often easy to solve (most of them are minor revisions asking some precisions on some points). I hope that you will agree with these proposals and that we will be able to publish your paper soon.

I would add 3 points.

1) In the review 2, you will find the request to put more attention (small discussion) about urban/rural definition (point 10 of his review), it is essential to refer to the EU+UN-Habitat+OECD work on urban/rural delineations on which I guess you based your typology?

- Applying the Degree of Urbanisation — A methodological manual to define cities, towns and rural areas for international comparisons — 2021 edition, DOI: 10.2785/706535

A paragraph on this discussion would be appropriate as it is fundamental in your model.

For a comparison of different approaches of "what is a city?", I would suggest a synthesis that I made recently (But I would not want to oblige you to quote my own work):

- Rozenblat C. (2020). Extending the concept of city for the delineation of large urban regions (LUR) for the cities of the world, Cybergeo, https://doi.org/10.4000/cybergeo.35411

2) for the approach on mobility: I think that you should distinguish the "local mobility" that you consider, to the "global" connectedness that could be evaluated by the air passengers flows of airports. I think that urban areas are also previously affected because they are highly globally connected. For this discussion I woudl suggest:

- Balcan, D., Colizza, V., Gonçalves, B., Hu, H., Ramasco, J. J., & Vespignani, A. (2009). Multiscale mobility networks and the spatial spreading of infectious diseases. Proceedings of the National Academy of Sciences, 106(51), 21484-21489.

3) As a spreading infection, the discussion could interprete also (with some limits) the COVID diffusion processes from urban to semi-urban and rural areas.

Journal Requirements:

Reviewers' comments:

Reviewer's Responses to Questions

**Comments to the Author**

1. Is the manuscript technically sound, and do the data support the conclusions?

Reviewer #1: Yes

Reviewer #2: No

2. Has the statistical analysis been performed appropriately and rigorously? 

Reviewer #1: Yes

Reviewer #2: No

3. Have the authors made all data underlying the findings in their manuscript fully available?

Reviewer #1: No

Reviewer #2: No

4. Is the manuscript presented in an intelligible fashion and written in standard English?

Reviewer #1: Yes

Reviewer #2: Yes

5. Review Comments to the Author

Reviewer #1: This paper studies determinants of COVID-19 spread at a regional level in Europe, and in a lesser extent gives descriptive statistics for US counties. Results are interesting regarding the respective role of density and mobility, and are directly useful for policy making. The paper is well written and structured, and the analysis are technically sound. I suggest the paper to be published after minor revisions.

A few remarks:

- L46: are there other possible indicators to quantify epidemic spreading? More background on using the reproduction number could be given for readers not familiar with standards in epidemiology.

- L53: citations for both R packages are not provided; more details on the estimation procedure could also be given for more clarity - for example is there a time-window on which the reproduction number is computed? In this case is there some optimal window size, and which size is used here?

- A discussion on data quality and possible biases (for cases, mortality, and mobility) would be useful to add.

- Linear statistical models are used here; could non-linear relationships be also considered?

- The results comparing Europe and US contrast with the rest of the results focusing on Europe which are more detailed - furthermore urban contexts and definitions of cities are quite different in Europe and the US, so the comparison may not be that straightforward. I would suggest to put US counties as supplementary material, and focus the paper on Europe with robust and comparable results.

- L241: using instrumental variable to disentangle causalities between mobility and reproduction number is suggested; more details could be provided, in particular which kind of variable could be used?

- L244: as lagged regression are used, Granger causality tests could be provided here, or a study of lagged correlations.

- Although mobility data can not be made publicly available for privacy reasons, source code should be provided (with synthetic mobility data for example) for replication on other case studies.

Reviewer #2: Summary:

The submitted paper aims to examine urban-rural gradients of COVID-19 spread and excess mortality. The authors first present data suggesting that the pandemic started earlier, spread faster, and was more deadly in more urban regions (at least for the first wave). They then show that mobility is generally higher in more urban areas and attempt to explain the increased spread and mortality as a result of this higher mobility.

Paper Strengths:

The broad coverage of geographical areas covered by the paper’s data are impressive and of general interest. In addition, the paper aims to highlight the nuance of infectious disease spread in different types of human settlements. These details are likely important to keep in mind as we continue to manage the spread of COVID-19, role out vaccination programs, and plan for future infectious disease outbreaks.

Paper Weaknesses:

Despite the strengths of this paper and the interest of the data it presents, there are major methodological concerns, clarity issues, and discrepancies between the data and conclusion that need to be addressed.

1) The authors should include a brief summary of the parameters used while calculating Rt. Because the authors aggregate to weekly averages, it is important for readers to understand that these are sliding windows.

2) The authors should justify their choice of 9 regression models and provide statistics to compare between models. If drawing conclusions from multiple models, multiple comparison correction should be employed.

3) The authors include a model called “Internal pca” which is not described in the caption or mentioned in the main text or methods.

4) The authors measure pandemic onset by the number o days between the beningng of 2020 and when the 20th confirmed case occurred. How was this number chosen? Are the results sensitive to different choices?

5) When comparing differences in Rt, mobility, and excess mortality, across urban-rural gradients, the authors should employ appropriate statistical tests and multiple comparison corrections (they certainly have enough data).

6) The methods section claims that the excess mortality time series spans from 2001 to 2015 but the results section claims that the baseline was calculated from data between 2011 and 2019. The authors should clarify and clearly state how the baseline was calculated (and with which data).

8) I would suggest the authors use log-population instead of population in their regression models as that is often a better indicator of ecological measures (e.g. Rt and mobility).

9) In the introduction the authors cite three papers which look at the relationship between city population and the spread of covid-19, and claim that these studies refer to population density. In fact, these studies are all based on urban scaling theory (Bettencourt, 2013) and specifically aim to understand the role of social network density in covid-19 spread. The distinction between social network density and population density is an important theoretical one and should be acknowledged.

10) Also in the introduction and the results it is crucial that the authors discuss the definitions of urban, intermediate, and rural regions being used. There are a number of different ways to decide what is a city (Taubenböck, 2012), and these choices might impact the interpretation of results.

11) There is inconsistency in how the terminology for the three mobility indicators. I would suggest sticking to a single choice for each indicator to improve clarity.

12) The authors should justify the comparison of models with different temporal shifts and put forth a hypothesis for how the results of these comparisons might indicate causality. As it stands, their data does not suggest any causal relationship between reproductive numbers and mobility. Part of this is that the methodology needs to be more clear.

13) The authors say that “mobility changes may respond endogenously to Rt values”. I question whether people actually respond to Rt values rather than perceived risk. It might be the case the people are checking Rt values on the internet and changing their behavior, but if so citations or additional evidence are needed.

14) Without proper statistics for model comparison, the discussion of the meaning of regression coefficients is difficult to justify. The authors should also address the fact that their regression models explain such a small proportion of the total variance in reproductive numbers.

15)

Figure 2 should have some sort of indication of the error in the calculated Rt values.

Figure 3 is confusing and the excess mortality axis should be color to match the bars.

Figure 5 is missing a description of what the color bar means.

Figure 6 is missing an x axis label.

16) The authors state that “mobility explains entirely the differences between the three groups of regions”. While it is true that differences in Rt by regional characterization are non-significant when conditioning on mobility, this statistical model treats regions with different levels of urbanization similarly. I suggest that the authors run an additional model where level of urbanization is treated as random effect to allow for different relationships between mobility and Rt in the different types of regions: a 50% reduction of mobility in a sparsely populated rural area may have only a very small effect of Rt, while a similar reduction in mobility might drastically impact spread in a large city.

Bibliography:

Bettencourt, L. M. (2013). The origins of scaling in cities. Science, 340(6139), 1438-1441.

Taubenböck, H., Esch, T., Felbier, A., Wiesner, M., Roth, A., & Dech, S. (2012). Monitoring urbanization in mega cities from space. Remote sensing of Environment, 117, 162-176.

6. PLOS authors have the option to publish the peer review history of their article (what does this mean?). If published, this will include your full peer review and any attached files.

Reviewer #1: No

Reviewer #2: No

---

## [Author Response · Author response to Decision Letter 0]

7 Jun 2021

We thank the editor and reviewers for the very useful suggestions provided. We provided detailed replies to each comment and results of sensitivity checks and additional analyses in the replies to reviewers document.

---

## [Decision Letter · Decision Letter 1]

6 Nov 2022

PONE-D-21-07940R1Territorial differences in the spread of COVID-19 in European regions and US countiesPLOS ONE

Dear Dr. Natale,

Thank you for submitting your manuscript to PLOS ONE. After careful consideration, we feel that it has merit but does not fully meet PLOS ONE’s publication criteria as it currently stands. Therefore, we invite you to submit a revised version of the manuscript that addresses the points raised during the review process.

We look forward to receiving your revised manuscript.

Kind regards,

Tzai-Hung Wen, Ph.D.

Academic Editor

PLOS ONE

Reviewers' comments:

Reviewer's Responses to Questions

**Comments to the Author**

1. If the authors have adequately addressed your comments raised in a previous round of review and you feel that this manuscript is now acceptable for publication, you may indicate that here to bypass the “Comments to the Author” section, enter your conflict of interest statement in the “Confidential to Editor” section, and submit your "Accept" recommendation.

Reviewer #3: All comments have been addressed

Reviewer #4: All comments have been addressed

2. Is the manuscript technically sound, and do the data support the conclusions?

Reviewer #3: Yes

Reviewer #4: (No Response)

3. Has the statistical analysis been performed appropriately and rigorously? 

Reviewer #3: Yes

Reviewer #4: (No Response)

4. Have the authors made all data underlying the findings in their manuscript fully available?

Reviewer #3: Yes

Reviewer #4: Yes

5. Is the manuscript presented in an intelligible fashion and written in standard English?

Reviewer #3: Yes

Reviewer #4: Yes

6. Review Comments to the Author

Reviewer #3: In the current work the authors study the territorial differences during the spread of COVID - 19 using indicators like Rt and the excess mortality for European and US counties. Then, they study the linear relation between Rt and mobility to confirm their hypothesis.

The paper is innovative, technically sound, the results strongly support the authors claims and the scientific problems were researched thoroughly. More specifically their hypothesis that mobility is a major factor which creates the territorial differences, is verified in the first wave.

In my opinion, this work should be accepted as is, as the authors revised the paper according to the previous reviewers’ comments and answered all the questions convincingly. Given that the produced results differ for different waves it would be interesting to observe the application of the same analysis to the 3rd or 4th wave. Moreover, another extension of this work would be to focus the study on the reasons of the differences between the waves.

Reviewer #4: The revised manuscript entitled “Territorial differences in the spread of COVID-19 in

European regions and US counties” evaluated transmission dynamics in Europe and the US. In general, the authors have responded all the comments from the previous reviewers. Below are some additional comments:

1. What is the motivation to compare the COVID-19 transmission in Europe and the US? What’s the definition of NUTS3 region? What is Rural-Urban Continuum Codes applied to the US? Although you indicated the detail in the footnote, I suggest giving brief description about this information.

2. What kind of regression you applied in the analysis?? The approach should be mentioned in the method but you described OLS and other detail in the result. Why do you include 3-week lags in the model?? Any scientific evidence to support the 3-week duration?

3. Although the title indicates the analysis will compare Europe and the US, it seems that the analysis about US only appeared in Figure 1 and 2. The excess mortality, mobility analysis only focused on Europe. The authors need to describe it clearly.

4. Table 1 and 2. What is the number in the parentheses of each coefficient? P-value?

5. The adjusted R-squared values in the regression models are very low. Are you able to draw reliable conclusions from the models??

7. PLOS authors have the option to publish the peer review history of their article (what does this mean?). If published, this will include your full peer review and any attached files.

Reviewer #3: No

Reviewer #4: No

---

## [Author Response · Author response to Decision Letter 1]

15 Nov 2022

Dear Editor,

the paper was subject to two full round of peer review and is awaiting publication since long. We believe that we have fully addressed the comments of the first two reviewers and, in this last round, also the points raised by reviewer 4. We thank you and all the anonymous reviewers for their time and interest in revising the manuscript.

Reviewer 3

In my opinion, this work should be accepted as is, as the authors revised the paper according to the previous reviewers’ comments and answered all the questions convincingly. Given that the produced results differ for different waves it would be interesting to observe the application of the same analysis to the 3rd or 4th wave. Moreover, another extension of this work would be to focus the study on the reasons of the differences between the waves. 

Response: The paper was originally submitted in March 2021 and has taken a long review process. During this long period the pandemic has progressed into 3rd or 4th waves. We have checked the mortality data for the EU and our findings about the lack of rural-urban differences are confirmed in the subsequent periods. As we indicate in the discussion major difference between first and second wave is linked to the widespread territorial diffusion of the infections following its initial onset in cities. We expect that similar effect continue to explain the rural-urban inversion in the later periods. We will consider the suggestions for further work.

Reviewer 4

1. What is the motivation to compare the COVID-19 transmission in Europe and the US? What’s the definition of NUTS3 region? What is Rural-Urban Continuum Codes applied to the US? Although you indicated the detail in the footnote, I suggest giving brief description about this information.RWe considered two areas of the world for which there was sufficiently detailed data on COVID19 cases at weekly and regional levels and with a harmonised classification allowing a comparison according to rural-urban typologies.

Response: We add further details on the classification of regions and rural-urban typologies in the EU and US.

2. What kind of regression you applied in the analysis?? The approach should be mentioned in the method but you described OLS and other detail in the result. 

Response: We provided more details on the regression and moved its descrition to the methods section. 

Why do you include 3-week lags in the model?? Any scientific evidence to support the 3-week duration?

Response: In the regression analyses, we use the mobility variables lagged by 3 weeks relative to the Rt value to account for the delay between the infection and the confirmation of the positive case. A sensitivity analysis of the lag is given in Figure 6. We selected a lag of 3 weeks in the past which is maximising the positive coefficient of mobility during the second wave. Positive lags produce as expected negative coefficients since mobility is reacting through restrictions rather than driving infection. We add more details to describe the choice of the lag in the text. Also, for Italy, Carteni et al. (2020) report that trips made three weeks earlier are the main determinants of new daily cases. 

With data on counties in the US between January and April 2020, Badr et al. (2020) show that declining mobility is strongly correlated with lower COVID-19 case growth rates, and that the impact may not be evident for up to three weeks. This is consistent with the incubation period of the first virus variants plus the additional time for reporting.

3. Although the title indicates the analysis will compare Europe and the US, it seems that the analysis about US only appeared in Figure 1 and 2. The excess mortality, mobility analysis only focused on Europe. The authors need to describe it clearly.

Response: We added a sentence clearly stating that the analysis of mobility covers only the EU due to the lack of sufficiently detailed mobility data for the US or other parts of the world.

4. Table 1 and 2. What is the number in the parentheses of each coefficient? P-value?

Response: We add in the caption of the tables a sentence specifying that numbers in parentheses represent the standard errors.

5. The adjusted R-squared values in the regression models are very low. Are you able to draw reliable conclusions from the models??

Response: Our purpose is not to predict the spread of the pandemic. For this we agree that there would be a need to achieve higher R-squared. The prediction of the spread of infections rather than simple OLS would require proper epidemiological models able to capture the typical temporal dynamics of epidemics. In our exercise we ignore these temporal dynamics and focus on the territorial differences and this explains the relatively low R-squared. The main aim of the paper is to see if the difference between rural and urban is significant and what is the role of mobility in explaining such difference. We tested alternative formulation of the model including as control variable a time dimension. This increasing dramatically the value of R-squared reflecting the time dependency in Rt values but has the downsize of cancelling and adsorbing most of the effect from our variables of interest.

---

## [Decision Letter · Decision Letter 2]

10 Jan 2023

Territorial differences in the spread of COVID-19 in European regions and US counties

PONE-D-21-07940R2

Dear Dr. Natale,

We’re pleased to inform you that your manuscript has been judged scientifically suitable for publication and will be formally accepted for publication once it meets all outstanding technical requirements.

Kind regards,

Tzai-Hung Wen, Ph.D.

Academic Editor

PLOS ONE

Additional Editor Comments (optional):

Reviewers' comments:

Reviewer's Responses to Questions

**Comments to the Author**

1. If the authors have adequately addressed your comments raised in a previous round of review and you feel that this manuscript is now acceptable for publication, you may indicate that here to bypass the “Comments to the Author” section, enter your conflict of interest statement in the “Confidential to Editor” section, and submit your "Accept" recommendation.

Reviewer #4: All comments have been addressed

2. Is the manuscript technically sound, and do the data support the conclusions?

Reviewer #4: Yes

3. Has the statistical analysis been performed appropriately and rigorously? 

Reviewer #4: Yes

4. Have the authors made all data underlying the findings in their manuscript fully available?

Reviewer #4: Yes

5. Is the manuscript presented in an intelligible fashion and written in standard English?

Reviewer #4: Yes

6. Review Comments to the Author

Reviewer #4: (No Response)

7. PLOS authors have the option to publish the peer review history of their article (what does this mean?). If published, this will include your full peer review and any attached files.

Reviewer #4: No

---

## [Editor Report · Acceptance letter]

13 Jan 2023

PONE-D-21-07940R2 

Territorial differences in the spread of COVID-19 in European regions and US counties 

Dear Dr. Natale:

I'm pleased to inform you that your manuscript has been deemed suitable for publication in PLOS ONE. Congratulations! Your manuscript is now with our production department. 

Kind regards, 

on behalf of

Dr. Tzai-Hung Wen 

Academic Editor

PLOS ONE